# QUANTIZED REINFORCEMENT LEARNING (QUARL)

## ABSTRACT

Recent work has shown that quantization can help reduce the memory, compute, and energy demands of deep neural networks without significantly harming their quality. However, whether these prior techniques, applied traditionally to image-based models, work with the same efficacy to the sequential decision making process in reinforcement learning remains an unanswered question. To address this void, we conduct the first comprehensive empirical study that quantifies the effects of quantization on various deep reinforcement learning policies with the intent to reduce their computational resource demands. We apply techniques such as post-training quantization and quantization aware training to a spectrum of reinforcement learning tasks (such as Pong, Breakout, BeamRider and more) and training algorithms (such as PPO, A2C, DDPG, and DQN). Across this spectrum of tasks and learning algorithms, we show that policies can be quantized to 6-8 bits of precision without loss of accuracy. We also show that certain tasks and reinforcement learning algorithms yield policies that are more difficult to quantize due to their effect of widening the models' distribution of weights and that quantization aware training consistently improves results over post-training quantization and oftentimes even over the full precision baseline. Additionally, we show that quantization aware training, like traditional regularizers, regularize models by increasing exploration during the training process. Finally, we demonstrate usefulness of quantization for reinforcement learning. We use half-precision training to train a Pong model 50% faster, and we deploy a quantized reinforcement learning based navigation policy to an embedded system, achieving an $18\times$ speedup and a $4\times$ reduction in memory usage over an unquantized policy.

## 1 INTRODUCTION

Deep reinforcement learning has promise in many applications, ranging from game playing (Silver et al., 2016; 2017; Kempka et al., 2016) to robotics (Lillicrap et al., 2015; Zhang et al., 2015) to locomotion and transportation (Arulkumaran et al., 2017; Kendall et al., 2018). However, the training and deployment of reinforcement learning models remain challenging. Training is expensive because of their computationally expensive demands for repeatedly performing the forward and backward propagation in neural network training. Deploying deep reinforcement learning (DRL) models is prohibitively expensive, if not even impossible, due to the resource constraints on embedded computing systems typically used for applications, such as robotics and drone navigation.

Quantization can be helpful in substantially reducing the memory, compute, and energy usage of deep learning models without significantly harming their quality (Han et al., 2015; Zhou et al., 2016; Han et al., 2016). However, it is unknown whether the same techniques carry over to reinforcement learning. Unlike models in supervised learning, the quality of a reinforcement learning policy depends on how effective it is in sequential decision making. Specifically, an agent's current input and decision heavily affect its future state and future actions; it is unclear how quantization affects the long-term decision making capability of reinforcement learning policies. Also, there are many different algorithms to train a reinforcement learning policy. Algorithms like actor-critic methods (A2C), deep-q networks (DQN), proximal policy optimization (PPO) and deep deterministic policy gradients (DDPG) are significantly different in their optimization goals and implementation details, and it is unclear whether quantization would be similarly effective across these algorithms. Finally, reinforcement learning policies are trained and applied to a wide range of environments, and it is unclear how quantization affects performance in tasks of differing complexity.

Here, we aim to understand quantization effects on deep reinforcement learning policies. We comprehensively benchmark the effects of quantization on policies trained by various reinforcement learning algorithms on different tasks, conducting in excess of 350 experiments to present representative and conclusive analysis. We perform experiments over 3 major axes: **(1)** environments (Atari Arcade, PyBullet, OpenAI Gym), **(2)** reinforcement learning training algorithms (Deep-Q Networks, Advantage Actor-Critic, Deep Deterministic Policy Gradients, Proximal Policy Optimization) and **(3)** quantization methods (post-training quantization, quantization aware training).

We show that quantization induces a regularization effect by increasing exploration during training. This motivates the use of quantization aware training, which we show demonstrates improved performance over post-training quantization and oftentimes even over the full precision baseline. Additionally, We show that deep reinforcement learning models can be quantized to 6-8 bits of precision without loss in quality. Furthermore, we analyze how each axis affects the final performance of the quantized model to develop insights into how to achieve better model quantization. Our results show that some tasks and training algorithms yield models that are more difficult to apply post-training quantization as they widen the spread of the models' weight distribution, yielding higher quantization error. To demonstrate the usefulness of quantization for deep reinforcement learning, we 1) use half precision ops to train a Pong model 50% faster than full precision training and 2) deploy a quantized reinforcement learning based navigation policy onto an embedded system and achieve an $18\times$ speedup and a $4\times$ reduction in memory usage over an unquantized policy.

## 2 RELATED WORK

Reducing neural network resource requirements is an active research topic. Techniques include quantization (Han et al., 2015; 2016; Zhu et al., 2016; Jacob et al., 2018; Lin et al., 2019; Polino et al., 2018; Sakr & Shanbhag, 2018), deep compression (Han et al., 2016), knowledge distillation (Hinton et al., 2015; Chen et al., 2017), sparsification (Han et al., 2016; Alford et al., 2018; Park et al., 2016; Louizos et al., 2018b; Bellec et al., 2017) and pruning (Alford et al., 2018; Molchanov et al., 2016; Li et al., 2016). These methods are employed because they compress to reduce storage and memory requirements as well as enable fast and efficient inference and training with specialized operations. We provide background for these motivations and describe the specific techniques that fall under these categories and motivate why quantization for reinforcement learning needs study.

**Compression for Memory and Storage:** Techniques such as quantization, pruning, sparsification, and distillation reduce the amount of storage and memory required by deep neural networks. These techniques are motivated by the need to train and deploy neural networks on memory-constrained environments (e.g., IoT or mobile). Broadly, quantization reduces the precision of network weights (Han et al., 2015; 2016; Zhu et al., 2016), pruning removes various layers and filters of a network (Alford et al., 2018; Molchanov et al., 2016), sparsification zeros out selective network values (Molchanov et al., 2016; Alford et al., 2018) and distillation compresses an ensemble of networks into one (Hinton et al., 2015; Chen et al., 2017). Various algorithms combining these core techniques have been proposed. For example, Deep Compression (Han et al., 2015) demonstrated that a combination of weight-sharing, pruning, and quantization might reduce storage requirements by 35-49x. Importantly, these methods achieve high compression rates at small losses in accuracy by exploiting the redundancy that is inherent within the neural networks.

**Fast and Efficient Inference/Training:** Methods like quantization, pruning, and sparsification may also be employed to improve the runtime of network inference and training as well as their energy consumption. Quantization reduces the precision of network weights and allows more efficient quantized operations to be used during training and deployment, for example, a "binary" GEMM (general matrix multiply) operation (Rastegari et al., 2016; Courbariaux et al., 2016). Pruning speeds up neural networks by removing layers or filters to reduce the overall amount of computation necessary to make predictions (Molchanov et al., 2016). Finally, Sparsification zeros out network weights and enables faster computation via specialized primitives like block-sparse matrix multiply (Ren et al., 2018). These techniques not only speed up neural networks but decrease energy consumption by requiring fewer floating-point operations.

**Quantization for Reinforcement Learning:** Prior work in quantization focuses mostly on quantizing image / supervised models. However, there are several key differences between these models and reinforcement learning policies: an agent's current input and decision affects its future state and actions, there are many complex algorithms (e.g: DQN, PPO, A2C, DDPG) for training, and there are many diverse tasks. To the best of our knowledge, this is the first work to apply and analyze the performance of quantization across a broad of reinforcement learning tasks and training algorithms.

## 3 QUANTIZED REINFORCEMENT LEARNING (QUARL)

We develop *QuaRL*, an open-source software framework that allows us to systematically apply traditional quantization methods to a broad spectrum of deep reinforcement learning models. We use the *QuaRL* framework to **1)** evaluate how effective quantization is at compressing reinforcement learning policies, **2)** analyze how quantization affects/is affected by the various environments and training algorithms in reinforcement learning and **3)** establish a standard on the performance of quantization techniques across various training algorithms and environments.

**Environments:** We evaluate quantized models on three different types of environments: OpenAI gym (Brockman et al., 2016), Atari Arcade Learning (Bellemare et al., 2012), and PyBullet (which is an open-source implementation of the MuJoCo). These environments consist of a variety of tasks, including CartPole, MountainCar, LunarLandar, Atari Games, Humanoid, etc. The complete list of environments used in the *QuaRL* framework is listed in Table 1. Evaluations across this spectrum of different tasks provide a robust benchmark on the performance of quantization applied to different reinforcement learning tasks.

**Training Algorithms:** We study quantization on four popular reinforcement learning algorithms, namely Advantage Actor-Critic (A2C) (Mnih et al., 2016), Deep Q-Network (DQN) (Mnih et al., 2013), Deep Deterministic Policy Gradients (DDPG) (Lillicrap et al., 2015) and Proximal Policy Optimization (PPO) (Schulman et al., 2017). Evaluating these standard reinforcement learning algorithms that are well established in the community allows us to explore whether quantization is similarly effective across different reinforcement learning algorithms.

**Quantization Methods:** We apply standard quantization techniques to deep reinforcement learning models. Our main approaches are post-training quantization and quantization aware training. We apply these methods to models trained in different environments by different reinforcement learning algorithms to broadly understand their performance. We describe how these methods are applied in the context of reinforcement learning below.

| Algorithm | OpenAI Gym | | Atari | | | | | | | PyBullet | | |
|---|---|---|---|---|---|---|---|---|---|---|---|---|
| | Cartpole | MountainCar (Continuous) | BeamRider | Breakout | MsPacman | Pong | Qbert | Seaquest | SpaceInvader | BipedalWalker | HalfCheetah | Walker2D |
| DQN | PTQ | n/a | PTQ | PTQ | PTQ | PTQ | PTQ | PTQ | PTQ | n/a | n/a | n/a |
| A2C | PTQ QAT BW | | PTQ QAT BW | PTQ QAT BW | PTQ QAT BW | PTQ QAT BW | PTQ QAT BW | PTQ QAT BW | PTQ QAT BW | | | |
| PPO | PTQ QAT BW | | PTQ QAT BW | PTQ QAT BW | PTQ QAT BW | PTQ QAT BW | PTQ QAT BW | PTQ QAT BW | PTQ QAT BW | | | |
| DDPG | n/a | PTQ | n/a | n/a | n/a | n/a | n/a | n/a | n/a | PTQ QAT BW | PTQ QAT BW | PTQ QAT BW |

Table 1: Summary of algorithms, environments, and quantization scheme in the *QuaRL* framework. PTQ means post-training quantization, QAT refers to Quantization-Aware Training, BW corresponds to evaluating the policy from 8-bits to 2-bits. The Atari games are the no frameskip versions with 4 frames stacked as input to the models. n/a means we cannot evaluate the combination due to algorithm-environment incompatibility. All put together, including the individual bitwidth experiments, we conduct over 350 experiments to present a deep understanding of how quantization affects deep reinforcement learning. This is the first such (comprehensive) study.

### 3.1 POST-TRAINING QUANTIZATION

Post-training quantization takes a trained full precision model (32-bit floating point) and quantizes its weights to lower precision values. We quantize weights down to fp16 (16-bit floating point) and int8 (8-bit integer) values. fp16 quantization is based on IEEE-754 floating point rounding and int8 quantization uses uniform affine quantization.

**Fp16 Quantization:** Fp16 quantization involves taking full precision (32-bit) values and mapping them to the nearest representable 16-bit float. The IEEE-754 standard specifies 16-bit floats with the format shown below. Bits are grouped to specify the value of the sign ($S$), fraction ($F$) and exponent ($E$) which are then combined with the following formula to yield the effective value of the float:

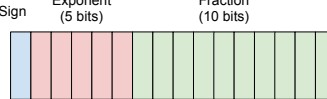

$$V_{fp16} = (-1)^S \times (1 + \frac{F}{2^{10}}) \times 2^{E-15}$$

In subsequent sections, we refer to float16 quantization using the following notation:

$$Q_{fp16}(W) = round_{fp16}(W)$$

**Uniform Affine Quantization:** Uniform affine quantization (TensorFlow, 2018b) is applied to a full precision weight matrix and is performed by 1) calculating the minimum and maximum values of the matrix and 2) dividing this range equally into $2^n$ representable values (where $n$ is the number of bits being quantized to). As each representable value is equally spaced across this range, the quantized value can be represented by an integer. More specifically, quantization from full precision to $n$-bit integers is given by:

$$Q_n(W) = \left\lfloor \frac{W}{\delta} \right\rfloor + z \text{ where } \delta = \frac{|min(W,0)| + |max(W,0)|}{2^n}, z = \left\lfloor \frac{-min(W,0)}{\delta} \right\rfloor$$

Note that $\delta$ is the gap between representable numbers and $z$ is an offset so that 0 is exactly representable. Further note that we use $min(W,0)$ and $max(W,0)$ to ensure that 0 is always represented. To dequantize we perform:

$$D(W_q, \delta, z) = \delta(W_q - z)$$

In the context of *QuaRL*, int8 and fp16 quantization are applied after training a full precision model on an environment, as per Algorithm 1. In post training quantization, uniform quantization is applied to each fully connected layer of the model (per-tensor quantization) and is applied to each channel of convolution weights (per-axis quantization); activations are not quantized. We use post-training quantization to quantize to fp16 and int8 values.

---

**Algorithm 1:** Post-Training Quantization for Reinforcement Learning

**Input:** $T$ : task or environment
**Input:** $L$ : reinforcement learning algorithm
**Input:** $A$ : model architecture
**Input:** $n$ : quantize bits (8 or 16)
**Output:** Reward
1 $M$ = Train($T, L, A$)
2 $Q = \begin{cases} Q_{int8} & n = 8 \\ Q_{fp16} & n = 16 \end{cases}$
3 **return Eval**($Q(M)$)

---

**Algorithm 2:** Quantization Aware Training for Reinforcement Learning

**Output:** Reward
**Input:** $T$ : task or environment
**Input:** $L$ : reinforcement learning algorithm
**Input:** $n$ : quantize bits
**Input:** $A$ : model architecture
**Input:** $Qd$ : quantization delay
1 $A_q$ = InsertAfterWeightsAndActivations($Q_n^{train}$)
2 $M, TensorMinMaxes$ = TrainNoQuantMonitorWeightsActivationsRanges($T, L, A_q, Qd$)
3 $M$ = TrainWithQuantization($T, L, M, TensorMinMaxes, Q_n^{train}$)
4 **return Eval**($M, Q_n^{train}, TensorMinMaxes$)

---

### 3.2 QUANTIZATION AWARE TRAINING

Quantization aware training involves retraining the reinforcement learning policies with weights and activations uniformly quantized to $n$ bit values. Importantly, weights are maintained in full fp32 precision except that they are passed through the uniform quantization function before being used in the forward pass. Because of this, the technique is also known as "fake quantization" (TensorFlow, 2018b). Additionally, to improve training there is an additional parameter, quantization delay (TensorFlow, 2018a), which specifies the number of full precision training steps before enabling quantization. When the number of steps is less than the quantization delay parameter, the minimum and maximum values of weights and activations are actively monitored. Afterwards, the previously

captured minimum and maximum values are used to quantize the tensors (these values remain static from then on). Specifically:

$$Q_n^{train}(W, V_{min}, V_{max}) = \left\lfloor \frac{W}{\delta} \right\rfloor + z \text{ where } \delta = \frac{|V_{min}| + |V_{max}|}{2^n}, z = \left\lfloor \frac{-V_{min}}{\delta} \right\rfloor$$

Where $V_{min}$ and $V_{max}$ are the monitored minimum and maximum values of the tensor (expanding $V_{min}$ and $V_{max}$ to include 0 if necessary). Intuitively, the expectation is that the training process eventually learns to account for the quantization error, yielding a higher performing quantized model. Note that uniform quantization is applied to fully connected weights in the model (per-tensor quantization) and to each channel for convolution weights (per-axis quantization). $n$ bit quantization is applied to each layer's weights and activations:

$$x_{k+1} = A(Q_n^{train}(W_k, V_{min}, V_{max})a_k + b) \text{ where } A \text{ is the activation function}$$

$$a_{k+1} = Q_n^{train}(x_{k+1}, V_{min}, V_{max})$$

During backward propagation, the gradient is passed through the quantization function unchanged (also known as the straight-through estimator (Hinton, 2012)), and the full precision weight matrix $W$ is optimized as follows:

$$\Delta_W Q_n^{train}(W, V_{min}, V_{max}) = I$$

In context of the *QuaRL* framework, the policy neural network is retrained from scratch after inserting the quantization functions between weights and activations (all else being equal). At evaluation full precision weights are passed through the uniform affine quantizer to simulate quantization error during inference. Algorithm 2 describes how quantization aware training is applied in *QuaRL*.

## 4 RESULTS

In this section, we first show that quantization has regularization effect on reinforcement learning algorithms and can boost exploration. Secondly, We show that reinforcement learning algorithms can be quantized safely without significantly affecting the rewards. To that end, we perform evaluations across the three principal axes of *QuaRL*: environments, training algorithms, and quantization methods.For post-training quantization, we evaluate each policy for 100 episodes and average the rewards. For Quantization Aware Training (QAT), we train atleast three policies and report the mean rewards over hundred evaluations. Table 1 lists the space of the evaluations explored.

**Quantization as Regularization:** To further establish the effects of quantization during training, we compare quantization-aware training with traditional regularization techniques (specifically layer-norm (Ba et al., 2016; Kukacka et al., 2017)) and measure the amount of exploration these techniques induce. It has been show in previous literature (Farebrother et al., 2018; Cobbe et al., 2018) that regularization actively helps reinforcement learning training generalize better; here we further reinforce this notion and additionally establish a relationship between quantization, generalization and exploration. We use the variance in action distribution produced by the model as a proxy for exploration: intuitively, since the policy samples from this distribution when performing an action, a policy that produces an action distribution with high variance is less likely to explore different states. Conversely, a low variance action distribution indicates high exploration as the policy is more likely to take a different action than the highest scoring one.

We measure the variance in action distribution produced by differently trained models (QAT-2, QAT-4, QAT-6, QAT-8, with layer norm and full precision) at different stages of the training process. We collect model rewards and the action distribution variance over several rollouts with deterministic action selection (model performs the highest scoring action). Importantly, we make sure to use deterministic action selection to ensure that the states reached are similar to the the distribution seen by the model during training. To separate signal from noise, we furthermore smooth the action variances with a smoothing factor of .95 for both rewards and action variances.

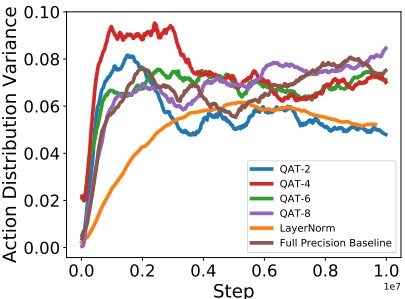 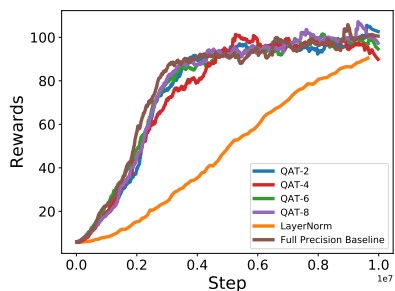

Figure 1: Exploration with different training processes and rewards achieved by corresponding models. Lower variance in inferred action distribution implies higher exploration. Training with higher quantization levels, like layer norm regularization, induces lower action distribution variance and thus higher exploration. Reward plot indicates that training with quantization achieves a similar level of rewards despite more exploration. Note that quantization during training is turned on after 5,000,000 steps (quant delay = 5,000,000) and the differences manifest shortly after this point.

Figure 4 shows the variance in action distribution produced by the models at different stages of training. Training with higher quantization levels (e.g: 2 bit vs 4 bit training), like layer norm regularization, induces lower action distribution variance and hence indicates more exploration. Furthermore, figure 4 reward plot shows that despite lower action variance, models trained with quantization achieve a reward similar to the full precision baseline, which indicates that higher exploration is facilitated by quantization and not by a lack of training. Note that quantization is turned on at 5,000,000 steps and we see its effects on the action distribution variance shortly after this point. In summary, data shows that training with quantization, like traditional regularization, in part regularizes reinforcement learning training by facilitating exploration during the training process.

**Effectiveness of Quantization:** To evaluate the overall effectiveness of quantization for deep reinforcement learning, we apply post-training quantization and quantization aware learning to a spectrum of tasks and record their performance. We present the reward results for post-training quantization in Table 2. We also compute the percentage error of the performance of the quantized policy relative to that of their corresponding full precision baselines ($E_{fp16}$ and $E_{int8}$). Additionally, we report the mean of the errors across tasks for each of the training algorithms.

The absolute mean of 8-bit and 16-bit relative errors ranges between 2% and 5% (with the exception of DQN), which indicates that models may be quantized to 8/16 bit precision without much loss in quality. Interestingly, the overall performance difference between the 8-bit and 16-bit post-training quantization is minimal (with the exception of the DQN algorithm, for reasons we explain in Section 4). We believe this is because the policies weight distribution is narrow enough that 8 bits is able to capture the distribution of weights without much error. In a few cases, post-training quantization yields better scores than the full precision policy. We believe that quantization injected an amount of noise that was small enough to maintain a good policy and large enough to regularize model behavior; this supports some of the results seen by Louizos et al. (2018a); Bishop (1995); Hirose et al. (2018); see appendix for plots showing that there is a sweet spot for post-training quantization.

For quantization aware training, we train the policy with fake-quantization operations while maintaining the same model and hyperparameters (see Appendix). Figure 2 shows the results of quantization aware training on multiple environments and training algorithms to compress the policies down from 8-bits to 2-bits. Generally, the performance relative to the full precision baseline is maintained until 5/6-bit quantization, after which there is a drop in performance. Broadly, at 8-bits, we see no degradation in performance. From the data, we see that quantization aware training achieves higher rewards than post-training quantization and also sometimes outperforms the full precision baseline.

| Algorithm → | A2C | | | | | DQN | | | | | PPO | | | | | DDPG | | | | |
|---|---|---|---|---|---|---|---|---|---|---|---|---|---|---|---|---|---|---|---|---|
| Datatype → | fp32 | fp16 | | int8 | | fp32 | fp16 | | int8 | | fp32 | fp16 | | int8 | | fp32 | fp16 | | int8 | |
| Environment ↓ | Rwd | Rwd | $E_{fp16}$ (%) | Rwd | $E_{int8}$ (%) | Rwd | Rwd | $E_{fp16}$ (%) | Rwd | $E_{int8}$ (%) | Rwd | Rwd | $E_{fp16}$ (%) | Rwd | $E_{int8}$ (%) | Rwd | Rwd | $E_{fp16}$ (%) | Rwd | $E_{int8}$ (%) |
| Breakout | 379 | 371 | 2.11 | 350 | 7.65 | 214 | 217 | -1.40 | 78 | 63.55 | 400 | 400 | 0.00 | 368 | 8.00 | | | | | |
| SpaceInvaders | 717 | 667 | 6.97 | 634 | 11.56 | 586 | 625 | -6.66 | 509 | 13.14 | 698 | 662 | 5.16 | 684 | 2.01 | | | | | |
| BeamRider | 3087 | 3060 | 0.87 | 2793 | 9.52 | 925 | 823 | 11.03 | 721 | 22.05 | 1655 | 1820 | -9.97 | 1697 | -2.54 | | | | | |
| MsPacman | 1915 | 1915 | 0.00 | 2045 | -6.79 | 1433 | 1429 | 0.28 | 2024 | -41.24 | 1735 | 1735 | 0.00 | 1845 | -6.34 | | | | | |
| Qbert | 5002 | 5002 | 0.00 | 5611 | -12.18 | 641 | 641 | 0.00 | 616 | 3.90 | 15010 | 15010 | 0.00 | 14425 | 3.90 | | | | | |
| Seaquest | 782 | 756 | 3.32 | 753 | 3.71 | 1709 | 1885 | -10.30 | 1582 | 7.43 | 1782 | 1784 | -0.11 | 1795 | -0.73 | | | | | |
| CartPole | 500 | 500 | 0.00 | 500 | 0.00 | 500 | 500 | 0.00 | 500 | 0.00 | 500 | 500 | 0.00 | 500 | 0.00 | | | | | |
| Pong | 20 | 20 | 0.00 | 19 | 5.00 | 21 | 21 | 0.00 | 21 | 0.00 | 20 | 20 | 0.00 | 20 | 0.00 | | | | | |
| Walker2D | | | | | | | | | | | | | | | | 1890 | 1929 | -2.06 | 1866 | 1.27 |
| HalfCheetah | | | | | | | | | | | | | | | | 2553 | 2551 | 0.08 | 2473 | 3.13 |
| BipedalWalker | | | | | | | | | | | | | | | | 98 | 90 | 8.16 | 83 | 15.31 |
| MountainCar | | | | | | | | | | | | | | | | 92 | 92 | 0.00 | 92 | 0.00 |
| Mean | | | 1.66 | | 2.31 | | | -0.88 | | 8.60 | | | -0.62 | | 0.54 | | | 1.54 | | 4.93 |

Table 2: Post training quantization error for DQN, DDPG, PPO, and A2C algorithm. The "Rwd" column corresponds to the rewards. The negative error percentage means the quantized policy performed better than fp32 policy. We summarize the error in rewards using arithmetic mean.

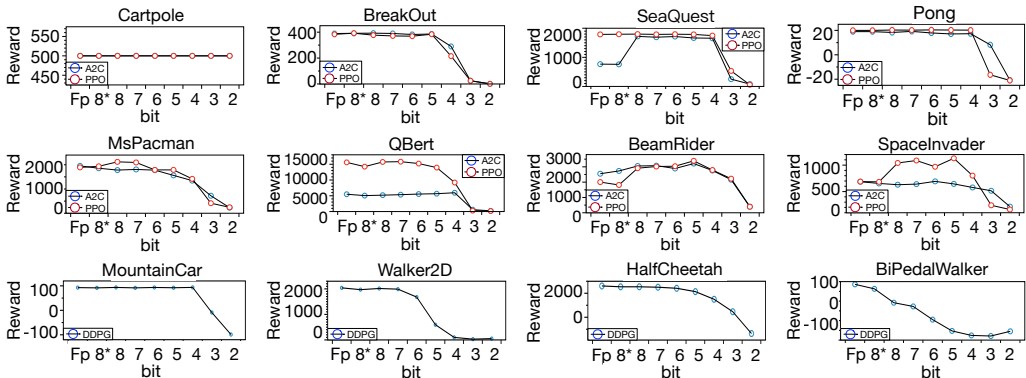

Figure 2: Quantization aware training (QAT) of PPO, A2C, and DDPG algorithms on OpenAI gym, Atari, and PyBullet. FP is achieved by fp32 and 8* is achieved by 8-bit post-training quantization.

**Effect of Environment on Quantization Quality:** To analyze the task's effect on quantization quality we plot the distribution of weights of full precision models trained in three environments (`Breakout`, `Beamrider` and `Pong`) and their error after applying 8-bit post-training quantization on them. Each model uses the same network architecture, is trained using the same algorithm (DQN) with the same hyperparameters (see Appendix).

Figure 3 shows that the task with the highest error (`Breakout`) has the widest weight distribution, the task with the second-highest error (`BeamRider`) has a narrower weight distribution, and the task with the lowest error (`Pong`) has the narrowest distribution. With an affine quantizer, quantizing a narrower distribution yields less error because the distribution can be captured at a fine granularity; conversely, a wider distribution requires larger gaps between representable numbers and thus increases quantization error. The trends indicate the environment affects models' weight distribution spread which affects quantization performance: specifically, environ-

| Environment | $E_{Int8}$ |
|---|---|
| Breakout | 63.55% |
| BeamRider | 22.05% |
| Pong | 0% |

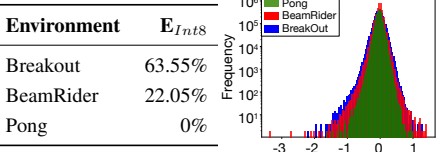

Figure 3: Weight distribution and corresponding 8-bit quantized error for models trained on the `Breakout`, `Beamrider` and `Pong` environments with DQN.

ments that yield a wider distribution of model weights are more difficult to apply quantization to. This observation suggests that regularizing the training process may yield better performance.

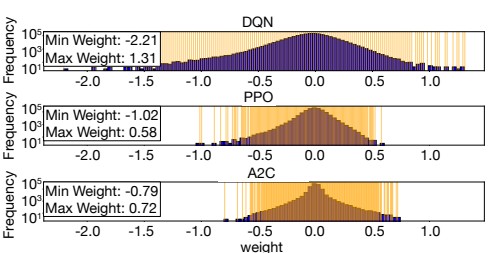

| Algorithm | Environment | fp32 Reward | $\mathbf{E}_{int8}$ | $\mathbf{E}_{fp16}$ |
|---|---|---|---|---|
| DQN | Breakout | 214 | 63.55% | -1.40% |
| PPO | Breakout | 400 | 8.00% | 0.00% |
| A2C | Breakout | 379 | 7.65% | 2.11% |

Table 3: Rewards for DQN, PPO, and A2C.

Figure 4: Weight distributions for the policies trained using DQN, PPO and A2C. DQN policy weights are more spread out and more difficult to cover effectively by 8-bit quantization (yellow lines). This explains the higher quantization error for DQN in Table 3. A negative error indicates that the quantized model outperformed the full precision baseline.

**Effect of Training Algorithm on Quantization Quality:** To determine the effects of the reinforcement learning training algorithm on the performance of quantized models, we compare the performance of post-training quantized models trained by various algorithms. Table 3 shows the error of different reinforcement learning algorithms and their corresponding 8-bit post-training quantization error for the Atari `Breakout` game. Results indicate that the A2C training algorithm is most conducive to int8 post-training quantization, followed by PPO2 and DQN. Interestingly, we see a sharp performance drop compared to the corresponding full precision baseline when applying 8-bit post-training quantization to models trained by DQN. At 8 bits, models trained by PPO2 and A2C have relative errors of 8% and 7.65%, whereas the model trained by DQN has an error of $\sim$64%. To understand this phenomenon, we plot the distribution of model weights trained by each algorithm, shown in Figure 4. The plot shows that the weight distribution of the model trained by DQN is significantly wider than those trained by PPO2 and A2C. A wider distribution of weights indicates a higher quantization error, which explains the large error of the 8-bit quantized DQN model. This also explains why using more bits (fp16) is more effective for the model trained by DQN (which reduces error relative to the full precision baseline from $\sim$64% down to $\sim$-1.4%). These results signify that the choice of RL algorithms (on-policy vs off-policy) have different objective functions and hence can result in a completely different weight distribution. A wider distribution has more pronounced impact on the quantization error.

## 5 CASE STUDIES

To show the usefulness of our results, we use quantization to optimize the training and deployment of reinforcement learning policies. We 1) train a pong model $1.5\times$ faster by using mixed precision optimization and 2) deploy a quantized robot navigation model onto a resource constrained embedded system (RasPi-3b), demonstrating $4\times$ reduction in memory and an $18\times$ speedup in inference. Faster training time means running more experiments for the same time. Achieving speedup on resource-constrained devices enables deployment of the policies on real robots.

**Mixed/Half-Precision Training:** Motivated by that reinforcement learning training is robust to quantization error, we train three policies of increasing model complexity (`Policy A`, `Policy B`, and `Policy C`) using mixed precision training and compare its performance to that of full precision training (see Appendix for details). In mixed precision training, the policy weights, activations, and gradients are represented in fp16. A master copy of the weights are stored in full precision (fp32) and updates are made to it during backward pass (Micikevicius et al., 2017). We measure the runtime and convergence rate of both full precision and mixed precision training (see Appendix).

| Algorithm | Network Parameter | fp32 Runtime (min) | MP Runtime (min) | Speedup |
|---|---|---|---|---|
| | Policy A | 127 | 156 | 0.87× |
| DQN-Pong | Policy B | 179 | 172 | 1.04× |
| | Policy C | 391 | 242 | 1.61× |

Table 4: Mixed precision training for reinforcement learning.

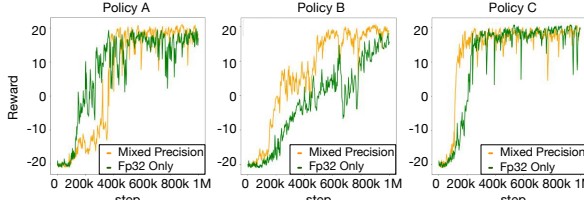

Figure 5: Mixed precision v/s fp32 training rewards.

Figure 5 shows that all three policies converge under full precision and mixed precision training. Interestingly, for `Policy B`, training with mixed precision yields faster convergence; we believe that

some amount of quantization error speeds up the training process. Table 5 shows the computational speedup to the training loop by using mixed precision training. While using mixed precision training on smaller networks (`Policy A`) may slow down training iterations (as overhead of doing fp32 to fp16 conversions outweigh the speedup of low precision ops), larger networks (`Policy C`) show up to a 60% speedup. Generally, our results show that mixed precision may speed up the training process by up to 1.6× without harming convergence.

**Quantized Policy for Deployment:** To show the benefits of quantization in deploying of reinforcement learning policies, we train multiple point-to-point navigation models (Policy I, II, and III) for aerial robots using Air Learning (Krishnan et al., 2019) and deploy them onto a RasPi-3b, a cost effective, general-purpose embedded processor. RasPi-3b is used as proxy for the compute platform for the aerial robot. Other platforms on aerial robots have similar characteristics. For each of these policies, we report the accuracies and inference speedups attained by the int8 and fp32 policies.

Table 5 shows the accuracies and inference speedups attained for each corresponding quantized policy. We see that quantizing smaller policies (Policy I) yield moderate inference speedups (1.18× for Policy I), while quantizing larger models (Policies II, III) can speed up inference by up to 18×. This speed up in policy III execution times results in speeding-up the generation of the hardware actuation commands from 5 Hz (fp32) to 90 Hz (int8). Note that in this experiment we quantize both weights and activations to 8-bit integers; quantized models exhibit a larger loss in accuracy as activations are more difficult to quantize without some form of calibration to determine the range to quantize activation values to (Choi et al., 2018).

A deeper investigation shows that Policies II and III take more memory than the total RAM capacity of the RasPi-3b, causing numerous accesses to swap memory (refer to Appendix) during inference (which is extremely slow). Quantizing these policies allow them to fit into the RasPi's RAM, eliminating accesses to swap and boosting performance by an order of magnitude. Figure 5 shows the memory usage while executing the quantized and unquantized version of Policy III, and shows how without quantization memory usage skyrockets above the total RAM capacity of the board.

| Policy Name | Network Parameters | fp32 (ms) | fp32 success (%) | int8 (ms) | int8 success (%) | Speed up |
|---|---|---|---|---|---|---|
| Policy I | 3L, MLP, 64 Nodes | 0.147 | 60% | 0.124 | 45% | 1.18 × |
| Policy II | 3L, MLP, 256 Nodes | 133.49 | 74% | 9.53 | 60% | 14 × |
| Policy III | 3L, MLP (4096, 512, 1024) | 208.115 | 86% | 11.036 | 75% | 18.85 × |

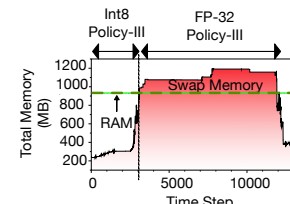

Figure 6: Table lists the inference speed in milliseconds (ms) on Ras-Pi3b+ and success rate (%) for three policies. The figure shows the memory consumption for Policy III's fp-32 and int8 policies.

In context of real-world deployment of an aerial (or any other type of) robot, a speedup in policy execution potentially translates to faster actuation commands to the aerial robot – which in turn implies faster and better responsiveness in a highly dynamic environment (Falanga et al., 2019). Our case study demonstrates how quantization can facilitate the deployment of a accurate policies trained using reinforcement learning onto a resource constrained platform.

# 6    CONCLUSION

We perform the first study of quantization effects on deep reinforcement learning using *QuaRL*, a software framework to benchmark and analyze the effects of quantization on various reinforcement learning tasks and algorithms. We analyze the performance in terms of rewards for post-training quantization and quantization aware training as applied to multiple reinforcement learning tasks and algorithms with the high level goal of reducing policies' resource requirements for efficient training and deployment. We broadly demonstrate that reinforcement learning models may be quantized down to 8/16 bits without loss of performance. Also, we link quantization performance to the distribution of models' weights, demonstrating that some reinforcement learning algorithms and tasks are more difficult to quantize due to their effect of widening the models' weight distribution. Additionally, we show that quantization during training acts as a regularizer which improve exploration. Finally, we apply our results to optimize the training and inference of reinforcement learning models, demonstrating a 50% training speedup for Pong using mixed precision optimization and up to a 18x inference speedup on a RasPi by quantizing a navigation policy. In summary, our findings

indicate that there is much potential for the future of quantization of deep reinforcement learning policies.

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

APPENDIX

Here, we list several details that are committed from the first 8 pages due to the limited page count. To the best of our ability, we provide sufficient details to reproduce our results and address common clarification questions.

## A  POST TRAINING QUANTIZATION RESULTS

Here we tabulate the post training quantization results listed in Table 2 into four separate tables for clarity. Each table corresponds to post training quantization results for a specific algorithm. Table 5 tabulates the post training quantization for A2C algorithm. Likewise, Table 6 tabulates the post training quantization results for DQN. Table 7 and Table 8 lists the post training quantization results for PPO and DDPG algorithms respectively.

| Environment | fp32 | fp16 | E_fp16 | int8 | E_int8 |
|---|---|---|---|---|---|
| *Breakout* | 379 | 371 | 2.11% | 350 | 7.65% |
| *SpaceInvaders* | 717 | 667 | 6.97% | 634 | 11.58% |
| *BeamRider* | 3087 | 3060 | 0.87% | 2793 | 9.52% |
| *MsPacman* | 1915 | 1915 | 0.00% | 2045 | -6.79% |
| *Qbert* | 5002 | 5002 | 0.00% | 5611 | -12.18% |
| *Seaquest* | 782 | 756 | 3.32% | 753 | 3.71% |
| *CartPole* | 500 | 500 | 0.00% | 500 | 0.00% |
| *Pong* | 20 | 20 | 0.00% | 19 | 5.00% |
| **Mean** | | | **1.66 %** | | **2.31 %** |

Table 5: A2C rewards for fp32, fp16, and int8 policies.

| Environment | fp32 | fp16 | E_fp16 | int8 | E_int8 |
|---|---|---|---|---|---|
| *Breakout* | 214 | 217 | -1.40% | 78 | 63.55% |
| *SpaceInvaders* | 586 | 625 | -6.66% | 509 | 13.14% |
| *BeamRider* | 925 | 823 | 11.03% | 721 | 22.05% |
| *MsPacman* | 1433 | 1429 | 0.28% | 2024 | -41.24% |
| *Qbert* | 641 | 641 | 0.00% | 616 | 3.90% |
| *Seaquest* | 1709 | 1885 | -10.30% | 1582 | 7.43% |
| *CartPole* | 500 | 500 | 0.00% | 500 | 0.00% |
| *Pong* | 21 | 21 | 0.00% | 21 | 0.00% |
| **Mean** | | | **-0.88%** | | **8.60%** |

Table 6: DQN rewards for fp32, fp16, and int8 policies.

| Environment | fp32 | fp16 | E_fp16 | int8 | E_int8 |
|---|---|---|---|---|---|
| *Breakout* | 400 | 400 | 0.00% | 368 | 8.00% |
| *SpaceInvaders* | 698 | 662 | 5.16% | 684 | 2.01% |
| *BeamRider* | 1655 | 1820 | -9.97% | 1697 | -2.54% |
| *MsPacman* | 1735 | 1735 | 0.00% | 1845 | -6.34% |
| *Qbert* | 15010 | 15010 | 0.00% | 14425 | 3.90% |
| *Seaquest* | 1782 | 1784 | -0.11% | 1795 | -0.73% |
| *CartPole* | 500 | 500 | 0.00% | 500 | 0.00% |
| *Pong* | 20 | 20 | 0.00% | 20 | 0.00% |
| **Mean** | | | **8.6%** | | **0.54%** |

Table 7: PPO rewards for fp32, fp16, and int8 policies.

## B  DQN HYPERPARAMETERS FOR ATARI

For all Atari games in the results section we use a standard 3 Layer Conv (128) + 128 FC. Hyperparameters are listed in Table 9.
We use stable-baselines (Hill et al., 2018) for all the reinforcement learning experiments. We use Tensorflow version 1.14 as the machine learning backend.

## C  MIXED PRECISION HYPERPARAMETERS

In mixed precision training, we used three policies namely Policy A, Policy B and Policy C respectively. The policy architecture for these policies are tabulated in Table 10.
For measuring the runtimes for fp32 adn fp16 training, we use the `time` Linux command for each run and add the `usr` and `sys` times to measure the runtimes for both mixed-precision training and fp32 training. The hyperparameters used for training DQN-Pong agent is listed in Table 9.

| Environment | fp32 | fp16 | E_fp16 | int8 | E_int8 |
|---|---|---|---|---|---|
| Walker2D | 1890 | 1929 | -2.06% | 1866 | 1.27% |
| HalfCheetah | 2553 | 2551 | 0.08% | 2473 | 3.13% |
| BipedalWalker | 98 | 90 | 8.16% | 83 | 15.31% |
| MountainCarContinuous | 92 | 92 | 0.00% | 92 | 0.00% |
| **Mean** | | | **1.54%** | | **4.93%** |

Table 8: DDPG rewards for fp32, fp16, and int8 policies.

| Hyperparameter | Value |
|---|---|
| n_timesteps | 1 Million Steps |
| buffer_size | 10000 |
| learning_rate | 0.0001 |
| warm_up | 10000 |
| quant_delay | 500000 |
| target_network_update_frequency | 1000 |
| exploration_final_eps | 0.01 |
| exploration_fraction | 0.1 |
| prioritized_replay_alpha | 0.6 |
| prioritized_replay | True |

Table 9: Hyper parameters used for mixed precision training for training DQN algorithm in all the Atari arcade learning environments.

## D  QUANTIZED POLICY DEPLOYMENT

Here we describe the methodology used to train a point to point navigation policy in Air Learning and deploy it on an embedded compute platform such as Ras-Pi 3b+. Air Learning is an AI research platform that provides infrastructure components and tools to train a fully functional reinforcement learning policies for aerial robots. In simple environments like OpenAI gym, Atari the training and inference happens in the same environment without any randomization. In contrast to these environments, Air Learning allows us to randomize various environmental parameters such as such as arena size, number of obstacles, goal position etc.

In this study, we fix the arena size to 25 m $\times$ 25 m $\times$ 20 m. The maximum number of obstacles at anytime would be anywhere between one to five and is chosen randmonly on episode to episode basis. The position of these obstacles and end point (goal) are also changed every episode. We train the aerial robot to reach the end point using DQN algorithm. The input to the policy is sensor mounted on the drone along with IMU measurements. The output of the policy is one among the 25 actions with different velocity and yaw rates. The reward function we use in this study is defined based on the following equation:

$$r = 1000 * \alpha - 100 * \beta - D_g - D_c * \delta - 1 \qquad (1)$$

Here, $\alpha$ is a binary variable whose value is '1' if the agent reaches the goal else its value is '0'. $\beta$ is a binary variable which is set to '1' if the aerial robot collides with any obstacle or runs out of the maximum allocated steps for an episode.[1] Otherwise, $\beta$ is '0', effectively penalizing the agent for hitting an obstacle or not reaching the end point in time. $D_g$ is the distance to the end point from the agent's current location, motivating the agent to move closer to the goal. $D_c$ is the distance correction which is applied to penalize the agent if it chooses actions which speed up the agent away from the goal. The distance correction term is defined as follows:

$$D_c = (V_{max} - V_{now}) * t_{max} \qquad (2)$$

$V_{max}$ is the maximum velocity possible for the agent which for DQN is fixed at 2.5 $m/s$. $V_{now}$ is the current velocity of the agent and $t_{max}$ is the duration of the actuation.

We train three policies namely Policy I, Policy II, and Policy III. Each policy is learned through curriculum learning where we make the end goal farther away as the training progresses. We terminate the training once the agent has finished 1 Million steps. We evaluate the all the three policies in fp32 and quantized int8 data types for 100 evaluations in airlearning and report the success rate.

---

[1]We set the maximum allowed steps in an episode as 750. This is to make sure the agent finds the end-point (goal) within some finite amount of steps.

| Algorithm | Policy Architecture |
|-----------|---------------------|
| Policy A | 3 Layer Conv (128 Filters) + FC (128) |
| Policy B | 3 Layer Conv (512 Filters) + FC(512) |
| Policy C | 3 Layer Conv (1024 Filters) + FC (2048) |

Table 10: The policy architecture that was used in mixed precision training for training DQN algorithm in Atari `Pong` environment.

We also take these policies and characterize the system performance on a Ras-pi 3b platform. Ras-Pi 3b is a proxy for the compute platform available on the aerial robot. The hardware specification for Ras-Pi 3b is shown in Table 11.

| Embedded System | Ras-Pi 3b |
|-----------------|-----------|
| CPU Cores | 4 Cores (ARM A53) |
| CPU Frequency | 1.2 GHz |
| GPU | None |
| Power | <1W |
| Cost | $35 |

Table 11: Specification of Ras-Pi 3b embedded computing platform. Ras-Pi 3b is a proxy for the on-board compute platform available in the aerial robot.

We allocate a region of storage space as swap memory. It is the region of memory allocated in disk that is used when system memory is utilized fully by a process. In Ras-Pi 3b, the swap memory is allocated in Flash storage.

E    POST-TRAINING QUANTIZATION SWEET SPOT

Figures 7 shows that there is a sweet spot for post-training quantization. Sometimes, quantizing to fewer bits outperforms higher precision quantization. Each plot was generated by applying post-training quantization to the full precision baselines and evaluating over 10 runs.

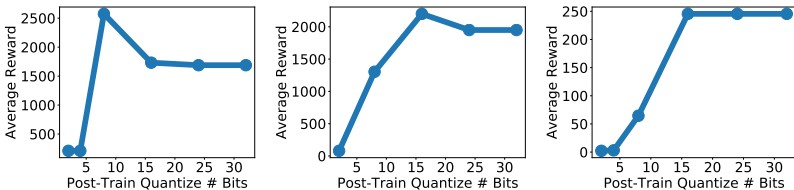

Figure 7: Post training quantization sweet spot for DQN MsPacman, DQN SeaQuest, DQN Breakout. We see that post-training quantization sweet spot depends on the specific task at hand. Note that 16-bit in this plot is 16-bit affine quantization, not fp16.

