# OpenReview forum: "Quantized Reinforcement Learning (QuaRL)"
_ICLR.cc/2020/Conference — Reject_

### Official Review · AnonReviewer2 · 2019-10-13
**Official Blind Review #2**

**Rating:** 3

**Review:**

This paper investigates the impact of using a reduced precision (i.e., quantization) in different deep reinforcement learning (DRL) algorithms. It shows that overall, reducing the precision of the neural network in DRL algorithms from 32 bits to 16 or 8 bits doesn't have much effect on the quality of the learned policy. It also shows how this quantization leads to a reduced memory cost and faster training and inference times.

I don't think this paper contributes with many novel results in the field, with most results being known or expected. The result that is interesting, in my opinion, is not properly explored.  The paper is well-written but it is a bit repetitive. It seems to me that the first 3 pages could be compressed in 1, as the same information is introduced over and over again.

With respect to the results being known, quantization is known to succeed in supervised learning tasks. In a deep reinforcement learning algorithm, when you apply post-training quantization in a deep reinforcement learning algorithm, mainly when that algorithm uses a value function (e.g., A2C or DQN), the problem is reduced to a regression problem. It is no different than a supervised learning problem. One has the original network’s prediction and they need to match that prediction. The complexities introduced in the reinforcement learning problem (bootstrapping, exploration, stability) don’t exist anymore as they arise during training. Thus, it doesn’t seem to me that these results are novel or surprising. In a sense it is neat to see that eventual errors do not compound, but that’s it. If I were to write this paper I would make this set of experiments much shorter just as a sanity check. One thing that I feel is missing is a notion of the impact of the quantization not in the rewards accumulated but in the policy/value function. How often does the quantized agent take a different action than the original agent, for example? Does it happen often but only when it doesn’t matter, or is it rare?

The quantization during training is potentially interesting. It was not properly explored though. I wonder if the quantization during training has a regularization effect, which is known to improve agent’s performance in reinforcement learning (e.g., Cobbe et al., 2018, Farebrother et al., 2018). Does the agent generalize better when using a network with fewer bits of precision? How does this change impact training? These are all questions that could potentially make the results in this paper novel (i.e., quantization as a form of regularization), but as it is now, the results are not that surprising.

Importantly, there are important details missing in the paper that make it hard for me to evaluate the validity of the results presented. Are the results reported over multiple runs? What is the version of the Atari games used, is it the one with stochasticity? How much variance do we have if we replicate this process over different networks that perform well? These are questions I would like to see answered because they also inform us about the impact of the proposed idea. For example, if by repeating this experiment multiple times one observe a high variance, it might mean that different models might be impacted in different ways.

The results in the “real-world” (Pong is not real-world) are not that surprising as well. Basically they show that if one uses a network with lower precision training and inference are faster, which, again, is not surprising.

There’s also an important distinction in the results that is not discussed in the paper: DQN estimates a value function while methods such as PPO directly estimate a policy. The reason DQN might have a wider distribution is exactly because it is estimating a different objective. These are important details that should be acknowledged and discussed in the paper. In my opinion, for this paper be relevant, it should have a very thorough evaluation of these different dimensions of reinforcement learning algorithms, with explicit discussions about it. Variance, the impact of quantization during learning, the distinction between parametrizing policies versus value functions, etc.

Finally, there are some aspects of the presentation of this paper that could also be improved. Aside from typos, below are some other comments on the presentation.
- There’s no such thing as Atari environment, it is either Arcade Learning Environment (Bellemare et al., 2013) or Atari games.
- I’d introduce/explain quantization in the beginning of the second paragraph of the Introduction for those not familiar with the term.
- No references are provided for the environments used. You should refer to Bellemare et al.’s (2013) work as well as Brockman et al.’s (2016).
- Is it really necessary to explain Fp16 quantization as it is done now, with even a picture of two bytes? I’d expect most readers are familiar with how numbers are represented in a computer.
- The equation for Uniform Affine Quantization is pretty much the same as the one in the Section Quantization Aware Training. All these “repetitions”, or discussions that are common-knowledge give the impression that the paper is trying to fill all the pages without necessarily having enough content.
- The references are not standardized (e.g., sometimes names are shortened, sometimes they are not) and the paper “Efficient inference engine on compressed deep neural network” is cited twice.


References:

Marc G. Bellemare, Yavar Naddaf, Joel Veness, Michael Bowling: The Arcade Learning Environment: An Evaluation Platform for General Agents. J. Artif. Intell. Res. 47: 253-279 (2013)

Greg Brockman, Vicki Cheung, Ludwig Pettersson, Jonas Schneider, John Schulman, Jie Tang, Wojciech Zaremba: OpenAI Gym. CoRR abs/1606.01540 (2016)


Karl Cobbe, Oleg Klimov, Christopher Hesse, Taehoon Kim, John Schulman: Quantifying Generalization in Reinforcement Learning. CoRR abs/1812.02341 (2018)

Jesse Farebrother, Marlos C. Machado, Michael Bowling: Generalization and Regularization in DQN. CoRR abs/1810.00123 (2018)


------


>>> Update after rebuttal: I stand by my score after the rebuttal.

The rebuttal did acknowledge some points I made to me the paper took a gradient update towards the right direction. I don't think the paper is quite there yet though. It is repetitive, spending too much time with basic concepts, and it still ignores small details that matter (e.g., calling it Atari Arcade Learning). I strongly recommend the authors to follow my recommendations closely and then submit the paper again to a next conference. The discussion about generalization is potentially interesting, going beyond the regularization for exploration aspect. A better discussion about quantization during learning is also essential. The first three pages could probably be compressed by half.

**Experience Assessment:**

I have published in this field for several years.

**Review Assessment: Checking Correctness Of Derivations And Theory:**

N/A

**Review Assessment: Checking Correctness Of Experiments:**

I carefully checked the experiments.

**Review Assessment: Thoroughness In Paper Reading:**

I read the paper thoroughly.

---

> ### Author Response · Authors · 2019-11-14
> **Response to Reviewer #2**
>
> Thank you for your thorough and insightful feedback. We especially appreciate your ideas on quantization as a form of regularization, which is now a core component in our paper and a  direction of major interest. We’ve summarized the main points below:
>
> Quantization as a form of regularization in RL-Training:
> Following the reviewer feedback, we’ve now established a relationship between using traditional regularizers (specifically layer-norm), quantization-aware training, and the amount of exploration/exploitation that an agent does (section 4). Specifically, we’ve found that quantization-aware training, like traditional regularizers, increases agent exploration and that more heavily quantized models tend to explore more.
>
> To quantify exploration, we look at the variance in the distribution of action values produced by the policy. Since during training an action is sampled from this distribution, if the policy produces a high variance action distribution (meaning it is much more confident in some actions over others), it is less likely to take other sets of actions and hence corresponds to lower exploration. Conversely, if the action distribution has low variance, the model is more likely to take one action versus another and hence corresponds to high exploration.
>
> We’ve plotted the variance in action distribution of different policies during different points in the training process and provide a more detailed analysis of the data in section 4 (Quantization as Regularization). Briefly, we found that
> Layer norm, a traditional form of regularization, reduces the variance in action distribution and hence increases exploration
> Quantization aware training likewise reduces the variance in action distribution and hence increases exploration
> Higher levels of quantization (e.g: quantization aware training at 2 bit vs 8 bit) increases exploration more than lower levels of quantization
> Quantization aware training yields policies that explore more, but attain the same levels of reward. Thus, action distribution variances are not lower because the model is less trained, but because the quantization actively facilitates exploration.
>
> Overall, the data indicates that quantization-aware training, like traditional regularizers (layer norm https://arxiv.org/abs/1710.10686) regularize RL training by increasing exploration.
>
> Post-training Quantization of reinforcement learning is same as supervised learning and thus it is not surprising that it works.
> [Authors]: We agree with the reviewer that post-training quantization in RL can be formulated as supervised learning. However, since reinforcement learning has a feedback loop, our work studies the effects of quantization on sequential decision and how the error compounds because of quantization. We quantitatively demonstrate that the error does not accumulate, which is an important finding for RL because if the error compounds then quantization benefits cannot be leveraged to improve system performance (Section 5).
>
> To show post training quantization applied to RL can improve system performance, we apply quantization to point-to-point navigation policy and achieve higher closed-loop control frequency on resource-constrained aerial robots. A higher control frequency translates to higher responsiveness and agility (http://rpg.ifi.uzh.ch/docs/RAL19_Falanga.pdf) of the aerial robot. Hence, understanding the effect of quantization w.r.t to compounding error is vital to exploit the system side benefits of quantization.
>
> The results in the “real-world” (Pong is not real-world) are not that surprising as well. Basically they show that if one uses a network with lower precision training and inference are faster, which, again, is not surprising.
>
> [Authors]:Originally, we used “real-world” to imply the usefulness of the approach but not the application (Pong). To avoid any confusion, we have updated the paper to remove any reference that suggests Pong is a real-world environment.
>
> One of our main contributions is to show that quantization applied to RL training can speed up the training process (by as much as 1.6X, Section 5) without impacting convergence. Speeding up the training can reduce the infrastructure cost thereby making it cheaper to train RL policies. We used Pong as a proxy RL environment to show the results. Given the promising results, we plan to use low-precision training to speed up RL training (e.g., robotics task covered in Section 5).
> How are the results reported? How many runs? What are the versions of the game?
> [Authors]: The results reported for post-training quantization are averaged over 100 episodes. The version of Atari games used is from gym specifically the NoFrameskip with 4 images stacked together. We updated the paper to with right attribution for the different environments used in this work.
>
> For quantization aware training, we train each policy at least 3 times and then for evaluating the reward from that policy we average it over 100 episodes.

---

### Official Review · AnonReviewer3 · 2019-10-23
**Official Blind Review #3**

**Rating:** 3

**Review:**

Training and deployment of DRL models is expensive. Quantization has proven useful in supervised learning, however it is yet to be tested thoroughly in DRL. This paper investigates whether quantization can be applied in DRL towards better resource usage (compute, energy) without harming the model quality. Both quantization-aware training (via fake quantization) and post-training quantization is investigated. The work demonstrates that policies can be reduced to 6-8 bits without quality loss. The paper indicates that quantization can indeed lower resource consumption without quality decline in realistic DRL tasks and for various algorithms.

The researchers propose a benchmark called QUARL that allows them to evaluate the effectiveness of quantization as well as the impact of quantization across a set of established DRL algorithms (e.g., DQN, DDPG, PPO) and environments (e.g., OpenAI Gym, ALE). Quantizations tested: fp32 -> fp16, int8, uniform affine.

The idea is simple and carries over from (image-based) supervised learning. The experiments are exhaustive and have to the best of my knowledge not yet been conducted. The conclusions indicate the advantage of quantization, however it is unclear how these results would generalize to real environments (the environments used are after all still simple benchmarks, e.g., half-cheetah or pong). The results are also not entirely surprising or impactful: how is quantization impacting reinforcement learning in a different way than supervised learning? E.g., DQN is supervised learning of a Q-value function against a target. What secondary effects does quantization have on the learning procedure: e.g., does it boost exploration behavior or does it regularize training? We also know that some of these tasks can be solved by extremely small models (https://arxiv.org/abs/1806.01363), while the models used in this work are significantly larger: is quantization working simply because the network capacity is large enough to allow it? These could be investigated in more detail. Furthermore, I'm also missing some experimental setup details: e.g., how many seeds were used for all of the experiments (which is known to greatly affect the results on the benchmarks used in this paper)?

**Experience Assessment:**

I have published in this field for several years.

**Review Assessment: Checking Correctness Of Derivations And Theory:**

I assessed the sensibility of the derivations and theory.

**Review Assessment: Checking Correctness Of Experiments:**

I assessed the sensibility of the experiments.

**Review Assessment: Thoroughness In Paper Reading:**

I read the paper at least twice and used my best judgement in assessing the paper.

---

> ### Author Response · Authors · 2019-11-14
> **Response to Reviewer #3**
>
> On behalf of all the authors, I would like to thank you for your feedback.
>
> It is unclear how results will generalize to real-environments (i.e Pong or Cheetah are not real world environments)
> [Authors] Thank you for the comment. We have updated the paper to remove any reference that suggest pong as a read-world environment. The purpose of evaluating on simple environments was to use it as baseline before moving onto complex tasks such as training RL policy for robot control. The details of applying quantization to RL policies are covered in Section V  (Quantized policy for deployment and Figure 5). In short, we show that one can increase the closed-loop frequency from 5Hz to 90Hz without much drop in the success rate.
>
> How is quantization affecting RL in a different way compared to supervised learning? For example does it boost exploration?
>
> We’ve established a relationship between using traditional regularizers (specifically layer-norm), quantization-aware training, and the amount of exploration/exploitation that an agent does. We’ve added a significant section on quantization and exploration in section 4 (Quantization as Regularization).
>
> In summary, we use the variance in the model’s inferred action distribution as a proxy for exploration. Higher variance in the action distribution implies less exploration as the model is biased to choosing one particular action during training. Our main additions are:
> Layer norm, a traditional form of regularization, reduces the variance in action distribution and hence increases exploration
> Quantization aware training likewise reduces the variance in action distribution and hence increases exploration
> Higher levels of quantization (e.g: quantization aware training at 2 bit vs 8 bit) increases exploration more than lower levels of quantization
> Quantization aware training yields policies that explore more, but attain the same levels of reward. Thus, action distribution variances are not lower because the model is less trained, but because the quantization actively facilitates exploration.
>
> Need information about experiments (number of runs).
> The results reported are averaged over 100 episodes. The version of Atari games used is from gym specifically the NoFrameskip with 4 images stacked together as inputs to the models. We use pybullet (Python API for Bullet Physics Engine) for half-cheetah, walker2d environments.

---

### Official Review · AnonReviewer1 · 2019-11-02
**Official Blind Review #1**

**Rating:** 3

**Review:**

This paper studies the effect of quantization on training reinforcement learning tasks. Specifically, the paper applies post-training quantization and quantization aware learning to various tasks and record the effects on accuracy and training speed.

Overall, the empirical evaluations suggest that quantization does not significantly hurt the performance of RL training among a wide range of tasks. On several tasks, the authors showed that quantization can significantly reduce memory usage and speed up the inference time. On the other hand, the improved efficiency comes at the cost of accuracy or lower rewards (2% - 5% error as shown in section 4) and (> 5% in terms of success rate as shown in Figure 5).

While it is expected that quantization should decrease the accuracy of the trained model, it is not entirely clear how one should evaluate the trade-off presented in the work. Some natural questions that I believe deserve more discussions are:
-- Are the kinds of accuracy cost the best one could hope for using these methods?
--  Is there still room for improvement in terms of reducing the cost of accuracy?

Detailed comments:
-- In the definition of Q_n(W): isn't $\delta$ equal to |W| / 2^n?
-- In Figure 5: your results show that the "int8" method has a significantly lower success rate than "fp32". Could you provide some discussion as to why this is the case?
-- Typos: Page 4, "is a applied"; Page 5, "full connected weights"; Page 8, "of a accurate".

**Experience Assessment:**

I do not know much about this area.

**Review Assessment: Checking Correctness Of Derivations And Theory:**

N/A

**Review Assessment: Checking Correctness Of Experiments:**

I assessed the sensibility of the experiments.

**Review Assessment: Thoroughness In Paper Reading:**

I made a quick assessment of this paper.

---

> ### Author Response · Authors · 2019-11-14
> **Response to Reviewer#1**
>
> On behalf of all the authors, I would like to thank you for your feedback.
>
> In the definition of Q_n(W): isn't delta equal to |W| / 2^n?
> [Authors:] In Q_n(w) , delta = (|max(W, 0)| + |min(W, 0)|) / 2^n. This effectively captures the range of the parameter weight values of W and divides this range into 2^n distinct segments. Dividing by delta effectively bins the unquantized values into these segments.
>
> In Figure 5: your results show that the "int8" method has a significantly lower success rate than "fp32". Could you provide some discussion as to why this is the case?
> [Authors:] Int8 shows a significantly lower success rate because we use post-training quantization to quantize both weights and activations. Importantly, quantization of activations is difficult since activations often exhibit a larger range of values (and the range of activations is unknown without doing forward passes on a representative sample of inputs). We’ve updated the paper with this information.
>
> Typos: Page 4, "is a applied"; Page 5, "full connected weights"; Page 8, "of a accurate"
> [Authors]: We have updated the paper and corrected typos.

---

### Decision · Program_Chairs · 2019-12-19

**Decision:**

Reject

**Comment:**

The paper investigates quantization for speeding up RL. While the reviewers agree that the idea is a good one (it should definitely help), they also have a number of concerns about the paper and presentation. In particular, the reviewers feel that the authors should have provided more insight into the challenges of quantization in RL and the tradeoffs involved. After having read the rebuttals, the reviewers believe that the authors are on the right track, but that the paper is still not ready for publication. If the authors take the reviewer comments and concerns seriously and update their paper accordingly, the reviewers believe that this could eventually result in a strong paper.